# Atypical Properties of a Thin Silver Layer Deposited on a Composite Textile Substrate

**DOI:** 10.3390/ma15051814

**Published:** 2022-02-28

**Authors:** Marcin Lebioda, Ewa Korzeniewska

**Affiliations:** Institute of Electrical Engineering Systems, Lodz University of Technology, Stefanowskiego 18, 90-537 Lodz, Poland; marcin.lebioda@p.lodz.pl

**Keywords:** thin films, thin layers, textronics, wearable electronics, joints

## Abstract

Thin layers are widely used in electronics and protective coatings. They are also increasingly used in wearable electronics. A major challenge affecting the use of thin layers is their connection to flexible substrates, particularly textile products. This article describes the stability of the resistance of a silver layer with a thickness of 250 nm in a wide temperature range of 15–295 K. The aim was to determine the temperature dependence of the resistance of layers formed on a composite textile substrate compared with that of layers produced on an Al_2_O_3_ substrate. The results showed that the electrical parameters of the layer formed on the composite textile substrate changed in a manner atypical for metallic layers. This may have been due to the polyurethane base layer. The roughness and ability to deform under the influence of heat of the substrate can significantly affect the electrical parameters of a thin metal layer produced by the PVD coating process, which is important for the design of textronic applications.

## 1. Introduction

Thin layers are used widely in engineering to improve the surface properties of solids, such as abrasion resistance, corrosion resistance, reflectance, hardness, and absorption [1]. They are also used to produce coatings with specific electrical properties [2,3], including electrically conductive layers for use in wearable electronics and textronics. Layer production methods include inkjet printing [4,5], magnetron sputtering [6], embroidery [7], weaving electrically conductive threads [8] or magnetic threads [9] into fabric structures, spin, spray or dip coating [10], and vacuum techniques such as CVD (chemical vapor deposition) [11] and PVD (physical vapor deposition) [12]. Several PVD processes can be distinguished: ion implantation or plating; ion beam mixing; plasma spraying or diffusion methods; cathodic sputtering; and high-vacuum evaporation. In the high-vacuum evaporation method, it is possible to use different evaporation sources, such as anodic or cathodic arc evaporators, induction or electron beam evaporators, and resistance-heated sources (indirectly or directly heated) [1].

The material from which the thin layer is made is evaporated in a high vacuum. The coating material is heated with the evaporation source to a high temperature until a sufficiently high vapor pressure is achieved to reach the desired evaporation rate. The rate of evaporation increases exponentially with the temperature of the source. A lower evaporation rate favors undesirable reactions with the tail gases, which may result in the formation of defective coatings containing particles of not only the coating material but its oxides. To keep coating contamination within acceptable limits, the residual gas pressure in the processing chamber must be sufficiently low during the coating process, and the material evaporation rate must be relatively slow. If the evaporation rate is too high, the particles of the evaporated material collide as a result of the excessive vapor pressure above the source. This causes some of the particles to return to the evaporation source [1]. If the temperatures of the evaporation sources are too high, this leads to the spontaneous formation of bubbles in the molten coating material. The material is then ejected by spattering from the evaporation source. The sprayed material not only settles in the vicinity of the vapor source but partially reaches the substrate, damaging the thin film [1]. The quality of the deposited layer can be assessed using optical microscopy or computed tomography [13,14]. The geometric shape of such layers can be changed using a laser beam [15].

As a result of the collision of vapor particles with the much colder surface of the substrate, the vapor particles move along the surface until they settle in a permanent place. The energy of the cohesion of the atoms of a pair is higher than the binding energy of the atom of the vapor with the substrate. Therefore, the vapor atom diffuses until it encounters another vapor atom and, as a consequence, forms a nucleus. In the initial phase, embryos are formed on defects in the substrate surface and generate islands. Subsequently, the islands grow until they merge and consequently form a continuous layer [1].

The method of forming a thin layer in the sputtering process and its morphology is described by Thornton [16,17,18]. In creating layers, four zones are distinguished depending on the ratio of the temperature of the substrate to the temperature of the source of metal evaporation. The layer morphology is as follows: porous columnar structure, porous grain structure, and recrystallized grain structure. In research [16], the essence of the pressure of working gas was pointed out from the perspective of the phenomenon of energy loss by atoms of evaporated and deposited material as a result of collision with gas molecules in the vacuum chamber during the layer formation process.

In the case of high pressure of the working gas, the atoms of the deposited material elastically collide with the sputtering gas atoms, and as a consequence, the vapor atom mean-free path is nearly the same as the source-to-substrate distance. For that reason, the deposition flux tilt component increases.

Furthermore, the mean kinetic energy of condensing particles is relatively small, and the mobility of the evaporated atoms is reduced. These factors influence the morphology of the structure, which is described as Zone I—porous columnar structure. In the case of low pressure of the working gas, the mobility of the atom increases, leading to the formation of a dense structure called Zone T, which is characterized by dense grain boundary arrays, high dislocation, and less roughness.

The working gas pressure does not affect the structure of the thin film, which can be characterized as a porous grain structure (Zone II) or a recrystallized grain structure (Zone III) [16,17,18].

Coatings are applied to various substrates, depending on the intended application. The task of the substrate is not only to provide mechanical support for the created thin layer but to ensure sufficient adhesion. The substrate should not react with the thin layer [19]. Alumina (Al_2_O_3_) is a cost-effective substrate with a wide range of applications for low- to medium-power DC and microwave circuits. For high-power DC and microwave circuits, aluminum nitride (AlN) is used. Beryllia (BeO) has extremely high thermal conductivity and is usually used for high-power DC or microwave circuits. When the device is based on a very smooth optical surface finish or contains micro- or millimeter-wave circuits requiring extremely low loss, thin films may be created on quartz or sapphire. Ferrite is used as a substrate for magnetically activated material [20]. Silicon is also commonly used for electronic devices [21,22]. For wearable electronics and textronics applications, a flexible composite textile material may be used as the substrate, which can be part of clothes or other wearable items [23].

It is known from research on the mechanical properties of thin metal layers that free, unsupported layers break under a tensile stress of 1–2%. According to Lacour et al. [24], thin metallic layers on rubber-like substrates stretched to twice their length did not lose their electrical conductivity. There have been no reports in the literature on the mechanical properties of metallic layers on textile substrates.

In addition to the substrate used for wearable electronics, it is important to select the connections between the textronic elements produced on textile substrates and the rest of the system. Gluing is the most often used method of connecting electric circuits to thin electrically conductive layers. Praveen and coworkers [25] described a method of connecting leads to thin Cu_2_O layers on an ITO layer deposited on a glass substrate. This structure was used to produce a gas sensor. The connections were made using a drop of electroconductive ink. To create a better joint, the surface to which the leads were glued was structured by making a set of very fine lines with a diamond pin (glass cutter), to remove the ITO layer on top of the lines. This procedure caused the formation of several electrically insulated stripes on the board, creating an uneven surface to which the connecting wires were glued. The quality of the connection between the thin-layer structure and the ITO was improved by evaporating gold on the thin layer in the form of a mesh. As a result of these procedures, the authors observed an increase in the sensitivity of the sensor.

Thin-film temperature sensors, such as those used in solid oxide fuel cells (SOFC), are highly sensitive to any disruptions in operation, limited space, and harsh operating conditions. Guk et al. [26] describe the complexity of connecting external line elements to complete an electric circuit. They proposed using wire connections based on a spring. They analyzed the influence of the contact resistance (R_c_) between the sensor pads and the attached cable on the temperature. The connection between the cable and thin layers was a technological problem, and the technique of fastening the cable, referred to as a spring-based connection (SBC), was found to be an effective method for applications that require transmission and control of an electric signal.

Wang and coworkers [27] took a different approach to connecting thin layers and electric signal conductors. Platinum jumper wires were attached to an ITO thin film structure by solidifying platinum paste, which was annealed at 850 °C for 30 min. Pt cables with insulation culverts were used to lead the signal to the resistance meter. The ITO layers were created by magnetron sputtering. The resistance of the metallic layers produced on aluminum substrates maintained a linear dependence on temperature up to about 600 °C [27]. However, it should be noted that when two conductive or semiconductive materials are connected on one side, voltage appears on the unpaired terminals if the ends are kept at different temperatures. This is known as the Seebeck effect [28]. The Seebeck effect is the basis of thermoelectric generation and the principle of operation of the thermocouple [29,30].

One trend observed in developing miniature electronics is wearable electronics, with particular emphasis on textronics. Connecting electrical circuits with textiles is a technological challenge. During the production of thin-film structures on textile substrates and in application research, we observed unusual changes in the resistance of the produced layers, not previously described in the literature. This article deals with the influence of glued, riveted, and soldered joints between thin-layer structures produced on a textile substrate, which can be easily combined with other textile products and other elements of electronic systems. This problem has not been described in the literature so far. The article also presents the results of research on atypical changes in the resistance of thin-film structures produced on a composite textile substrate in comparison with the changes in such layers produced on a ceramic substrate.

Here, we investigated the effect of connections made between connecting wires and thin layers made by physical vacuum deposition on a textile substrate. We examined the effect of the textile substrate on the resistance of the layers.

## 2. Materials and Methods

### 2.1. Sample Preparation

Thin layers were produced using the PVD (physical vacuum deposition) process on a commercially available composite textile substrate, trade name Cordura. Cordura’s high mechanical strength justified choosing it as the base material. The area density of the material is 460 g/cm^2^. It is made of nylon threads covered with a polyurethane layer, thanks to which it is possible to create a thin electrically conductive layer on the surface. Irregularities, which are a natural element as a consequence of weaving the thread, must be leveled to ensure that the textile substrate can be evened out. It is not possible to form a continuous layer with good electrical properties without an intermediate layer, such as one of polyurethane. Polyurethane additionally fills the spaces between the nylon fibers and limits the possibility of their movement during the material deformation. The roughness of the surface was assessed using optical computed tomography, as presented previously [31]. After cleaning with isopropyl alcohol, the support material was conditioned for 24 h at 20 °C. It was then placed in the chamber of a Classic 250 Pfeiffer Vacuum (Wetzlar, Germany) stand. Silver with a purity of 99.99% was used as the deposited metal. The deposition process was started at an initial vacuum of 0.05 Pa (5.0 × 10^−4^ mbar) and continued for 5 min [32]. Tungsten boats, which have a much higher melting point than evaporated silver, were used as a resistance source. The substrate was placed at a distance of 6 cm from the vapor source. During the process, the base on which the substrate material was placed was rotated in order to ensure the greatest possible uniformity of the thin layer.

The layer thickness was measured indirectly. Layers were applied to the textile substrate and to glass with the same process. The surface morphology of the textile material made it impossible to precisely determine the thickness of the layer. The layer on glass was treated as the reference and measured using a Dektak 3ST (Veeco Plainview, New York, NY, USA) profilometer. The average measurements showed that the thicknesses of the manufactured layers were in the range of 250 nm. This value was the average value of several measurements on a glass substrate, and the difference of the obtained results for the glass substrate did not exceed 20 nm. Direct measurement on a textile base was not possible because of the elasticity of the substrate and the contact measurement method. A sample of the textile substrate (Cordura) with dimensions of 35 × 5 mm was coated with Ag and joined with 35 μm Ag foil strips with dimensions of 40 × 5 mm (Figure 1)

Figure 1 shows information about the location of the thermocouples to measure the temperature of the sample.

A joint with dimensions of about 5 × 5 mm was made between a thin Ag layer deposited on the substrate and a 35 μm metal foil. The possibilities for making connections to a layer deposited on a textile substrate are very limited. We used the two most common methods of joining: gluing with electroconductive paste (sample G + T) [33] and crimping with brass rivets of inner diameter 2.5 mm (sample C + T). The glue used, ELPOX AX 15S (Amepox Ltd., Lodz, Poland), consists of two components based on epoxy resin. After mixing, the glue is a smooth paste of bright silver color. More technical data is available at the producer website [33]. Both methods ensured permanent and durable joints between the textiles and the metal current lead without significantly damaging the conducting Ag layer. A textile sample with additional electrodes made of Ag wires connected with electroconductive paste (sample M + T) was tested using the four-probe method. An Al_2_O_3_ ceramic substrate with the same dimensions as the textile samples was used as a reference. The silver layer was deposited on the textile or ceramic in a single sputtering process, ensuring the same thickness of the Ag layer. We used two methods of joining the Ag layer to the ceramic substrate: gluing (the same electroconductive paste) (sample G + C) and soldering (sample S + C). Crimping cannot be used for ceramic substrates, because it causes permanent damage to the brittle substrate. We used indium soldering, which ensures joints with very good electrical and mechanical properties [34]. All samples are presented in Table 1.

Samples with additional voltage electrodes were also prepared, in order to measure the voltage drop directly on the Ag layer. For this purpose, thin silver wires (diameter 150 µm) were glued directly to the Ag layer (Figure 1c).

### 2.2. Instrumentation and Measurement Procedure

The electrical parameters of the samples were measured at room temperature and at cryogenic temperature. The tests at cryogenic temperatures were performed to analyze the temperature dependence of the electrical resistance of the Ag layer deposited on the textile substrate. The results determined the conductivity of the layer compared with that of bulk Ag samples. We used rectangular 25 × 10 mm samples without permanent connections (glued or clamped) to eliminate possible joint effects. An HP34420A Micro-Ohm meter (Keysight Technologies, Santa Rosa, CA, USA) was used to measure the resistance of the samples by the four-probe method. The flat silver electrodes were integrated into the holder, which allowed the samples to be mounted on the copper heat exchanger in the cryostat (Figure 2).

Studies were performed using a helium closed-cycle DE-210 cryostat (Advanced Research Systems, Inc., Macungie, PA, USA) and a Lake Shore 331 temperature controller (Lakeshore Cryotronics Inc., Westerville, OH, USA). The samples were placed in a vacuum chamber, cyclically cooled, and heated over a temperature range of 15–300 K at a rate of about 4 K/min. A reference temperature sensor (silicone diode DT-670-SD Lakeshore Cryotronics Inc., Westerville, OH, USA) was mounted directly on the textile substrate. A massive copper heat exchanger with a holder and sample was attached to the “cold finger” of the cryocooler.

Measurements of the parameters of the samples at room temperature focused on analysis of the Joule heat flux density generated by the current in the samples. Generally, Joule heat changes the electrical resistance of the Ag layer and the mechanical properties of the substrate. To limit the influence of external factors, such as mechanical stress or uncontrolled heat transfer, the test samples were mounted in a special rack with thin threads supporting each sample (Figure 3). The special rack reduced heat exchange with the environment to natural convection along the entire length of the samples, as well as reducing heat conduction in the silver foil strips and supply wires. The sample was supported with thin threads of negligible thermal conductivity to minimize the mechanical stress on the sample without affecting the heat exchange with the environment.

The rack made it possible to mount thin thermocouples (T-type, wire diameter 140 μm), which measured the temperatures of the Ag layer on the textile substrate and of the layer-to-foil joint. We used foam board to make the rack. Foam board has low thermal conductivity and good mechanical stability. The samples were supplied from a controllable current source. The sample voltage drop was measured using an HP34420A digital multimeter (Keysight Technologies, Santa Rosa, CA, USA). The current was measured using an HP34401A digital multimeter (Keysight Technologies, Santa Rosa, CA, USA). The voltage drop was measured in two configurations: on the entire sample, including the joints, and on only part of the metallic layer. For this purpose, additional samples were prepared with Ag wire voltage electrodes glued to the Ag layer (Figure 1c).

## 3. Results and Discussion

We performed two types of tests of thin Ag layers on textile. In the first case, the electrical properties of the thin Ag layer applied to the composite substrate (Cordura) were tested quasistatically in a wide range of temperatures, including cryogenic temperatures. In this case, Joule heat was not generated in the film. The aim of the study at cryogenic temperatures was to determine the dependence of layer resistance on temperature without self-heating by Joule heat. The sample was cooled with low dynamics of temperature changes, and the high heat capacity of the copper exchanger eliminated temperature gradients in the sample. The second case consisted of a time-dependent study with Joule heat generated by an electric current in the thin layer.

Figure 4 shows the relative change in the resistance of the layer for a temperature range of 15–295 K compared with that of a pure bulk silver sample [35]. The characteristics of the thin layer were repeatable and continuous. There were no sudden, unpredictable changes in resistance. This proved the continuity of the metallic layer, good adhesion to the substrate, and a lack of propagation of structural defects related to temperature changes. The tests were performed for six samples (six pieces of textile substrate with deposited Ag layers). The differences in the results did not exceed 5%. The tested sample was mechanically pressed against the heat exchanger, which may have introduced additional stresses and slightly changed the course of the characteristics. The changes in resistance were typical for metals, wherein the phonon scattering mechanism predominates [35,36]. The nature of changes in *R*(*T*) for the layer and bulk Ag were very similar, and a flattening of the characteristics below 40 K was observed. This was attributed to the deactivation of the phonon scattering mechanism [35,36]. The relative change in resistance (Figure 4) was much smaller for the thin layer. For pure bulk silver, the dynamics of the changes were higher, which would be due to a lower concentration of defects [16,17,18]. In the 40–295 K range, the change in resistance was close to linear (Figure 4). The change in temperature coefficient of resistance for the thin layer (α_Ag_layer_ = 2.6∙10^−3^ 1/K) was much smaller than that for the bulk pure Ag (α_Ag_bulk_ = 3.8∙10^−3^ 1/K) [37,38,39]. This difference was a consequence of the heterogeneity of the thin layer and the occurrence of structural defects [16,17,18] resulting in part from the surface profile of the composite substrate and partly from the grain effect of the metallic layer.

The aim of studying the electrical properties of the thin layers at room temperature was to determine the dominant Joule heat source generated as a result of current transport through the sample (Figure 1) and the accompanying changes in the electrical parameters of the layer. Tests were carried out for samples with silver electrodes connected to the conductive layer by two methods: gluing and riveting. The resistance of the samples was in the range of 600–800 mΩ. Regardless of the electrode bonding method used, the results showed a change in the layer resistance uncharacteristic for metals as a result of Joule heat generation. It should be noted that the effect was observed only in the case of self-heating of the samples during the transport of electric current through the thin Ag layer.

Figure 5 shows the results of measurements of layer temperature changes (Figure 5a), the silver lead at the connection point (Figure 5b), and the voltage drop changes across the entire sample (Figure 5c). The changes in the temperature of the layer and the electrode were typical for a convection-cooled system maintaining natural convection. After the transition, which was a result of heating the sample, the temperature of the layer and the electrode stabilized, but the temperature of the layer was much higher. This was due mainly to its much higher layer resistance compared with that of the electrodes made of Ag strips. Moreover, the dissipation of heat from the Ag electrodes and the connection with the layer was much more effective. In the presented configuration, the Ag electrodes acted as heat sinks that dissipated heat to the environment. The substrate had much lower thermal conductivity than the electrodes and reduced the emission of heat to the environment by insulating the layer. Heat transfer in the Ag layer was very limited, because of the small thickness of the layer, so the heat flux from the layer to the electrodes was small. However, the steady-state temperature values of both the layer and the electrode were stable and invariant. The characteristics of *U* = f(*T*) (Figure 5d) for *I* ≥ 300 mA showed a slight increase in voltage in the initial stage of heating as a result of the increase in resistance typical for metals as the temperature increases. However, further heating caused an abnormal drop in the resistivity of the sample, because of which a reduction in voltage was observed across the sample. In the case of the samples with glued electrodes (Figure 5), the effect of the unusual decrease in resistance was less noticeable than in the case of the samples with riveted electrodes (Figure 6). In the latter case, the changes in resistance were much larger, and the initial stage of increasing resistance, associated with temperature increase, was not observed. For higher current values (*I* ≥ 200 mA), this effect was masked by the unusual reduction in layer resistance (Figure 6c). Figure 5d and Figure 6d compare the temperature and voltage changes in both types of samples for 500 mA current. The changes in voltage, regardless of the type of connection, correlated with the changes in temperature.

Glued wires were an additional heat load and changed the mechanical properties of the layer. Lower changes in temperatures were observed than in the samples with clamped electrodes. Connections made by clamping (riveting) had slightly higher resistance. This was mainly the effect of the smaller contact area between the Ag electrode (foil) and the thin layer applied to the composite substrate. The clamping method requires making holes in the substrate with the layer and in the electrode, which affects the connection surface. As a result, the resistance of the joint contributed to increase the Joule heat generated in the joint compared with that generated in the glued joint (Figure 7). However, clamping limited the dynamics of the temperature changes. The massive rivets also slightly reduced the temperature of the layer in the steady state.

The initial temperature of the samples was 21–23 °C (as can be seen in Figure 8). This meant that the temperature after heating the sample with Joule’s heat ranged from 23 to about 40 °C depending on the value of the flowing current (100–700 mA). The selection of the temperature range was not accidental and correlated with the typical differences between the temperature of the human body and the ambient temperature. The temperature distributions obtained by the contact method were confirmed by thermovision measurements (Figure 8). The sample with clamped electrodes showed a slight increase in the temperature of the connections (Figure 8a) compared with the sample with glued electrodes (Figure 8b).

The unusual changes in the resistance of the layer created on the textile substrate were directly related to the structure of the composite substrate (compare the plots in Figure 9 and Figure 10). As a consequence of thermal deformation of the polyurethane layer on which the thin layer of Ag was formed (Figure 9a), the structure of the composite deformed heterogeneously, which translated into deformation of the Ag layer. The complex surface morphology promoted changes in the layer resistance and could lead to unusual changes in resistance.

At the point of connection with the silver electrode (foil), deformation of the substrate and Ag layer was very limited by the rigid electroconductive paste layer and the strong adhesion of the paste to the substrate surface. In Figure 9b, a continuous layer of paste (Figure 9b-2) adhering to the Ag electrode (Figure 9b-1) can be seen, as can the textile substrate (Figure 9b-4) coated with a polyurethane layer (Figure 9b-3). The joint was also characterized by other thermal parameters resulting from its geometry and the thermal properties of the used materials (Ag foil + paste), which are not without significance for the process of heat transport in the volume of the tested samples.

To confirm the cause of the anomaly in the change of resistance, tests were carried out on a sample with additional voltage electrodes glued directly to the Ag layer (Figure 1c). The results (Figure 10) unequivocally showed that the change in the resistance of the sample was directly related to the changes taking place in the thin layer. The changes in temperature (Figure 10a,b) and voltage (Figure 10c), as well as the relationship between them (Figure 10d), were identical in shape to the measurements made for samples without voltage electrodes. However, the addition of supplementary electrodes (gluing) changed either the electrical or thermal parameters of the sample, making a correct quantitative comparison of the results impossible.

As part of the analysis of thermoelectric phenomena, tests were carried out on samples with the Ag layer deposited on a ceramic substrate (Figure 1d). The results (Figure 11) differed significantly from those for the composite substrate. The good thermal conductivity of the substrate determined the course of thermal processes, as a result of which the temperature changes of the layer and the electrode at the connection point were very similar (Figure 11a,b,d,e) regardless the method of connection. The change in resistance due to temperature increase, which can be seen in the form of a slight increase in voltage drop, is typical for metals (Figure 11c,f).

## 4. Conclusions

This article presents the results of research on the electrical properties of thin silver layers deposited by the PVD process on a textile composite substrate (Cordura). The first stage of the research included analysis of the influence of temperature on the layer resistance. Tests carried out in a wide temperature range (15–295 K) showed that the *R*(*T*) dependence was typical for metals, while the temperature coefficient of resistance (α_Ag_layer_ = 2.6∙10^−3^ 1/K) was lower than the value characteristic for massive Ag samples (α_Ag_bulk_ = 3.8∙10^−3^ 1/K). This was directly related to the structure of the Ag layer and the morphology of the substrate surface. The tested layer remained continuous and stable in the full temperature range.

Atypical properties were observed when we studied the transport of electric current through the layer and accompanying Joule heat generation. In a system where the layer was the dominant heat source, the layer’s electrical parameters changed in a way unusual for metallic layers. Despite stabilization of the temperature of the layer and the connection, the resistance of the layer decreased. The correctly designed and fabricated connections between the layer and the electrodes (Ag foil) did not have a significant qualitative or quantitative influence on the results. Our analysis of the phenomenon indicated that a component of the composite substrate—the polyurethane layer—was the source of the anomalous changes in layer resistance. It can be hypothesized that appropriate modification of the surface of the textile substrate and/or the addition of an intermediate layer would enable the electrical parameters of the applied metallic layer to be controlled. In the analyzed configuration, the properties of the layer were determined by the parameters of the substrate. This is a feature that may be of importance in the design and application of textronic systems. Appropriate local modification of the substrate in combination with one-step application of the metallic layer may simplify and speed up the process of producing structures with different properties. Future work will focus mainly on analyzing the electrical properties of metallic layers applied to other composite textile substrates and the possibility of modifying substrates to change the properties of the metallic layers.

## Figures and Tables

**Figure 1 materials-15-01814-f001:**
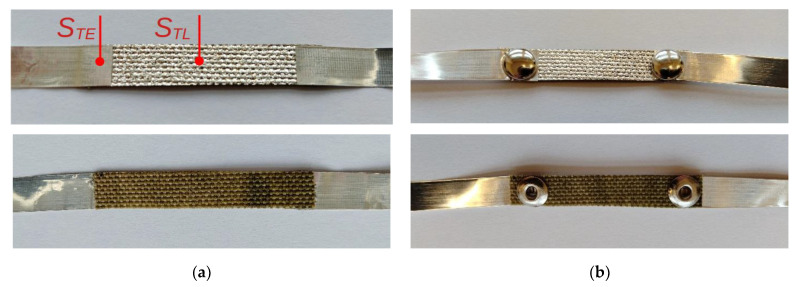
Front and back views of samples with current leads and temperature sensors: (**a**) gluing, S_TE_—position of the electrode temperature sensor (or connections), S_TL_—position of the layer temperature sensor (sample G + T); (**b**) crimping (sample C + T); (**c**) wire voltage electrodes (sample M + T); (**d**) sample S + C.

**Figure 2 materials-15-01814-f002:**
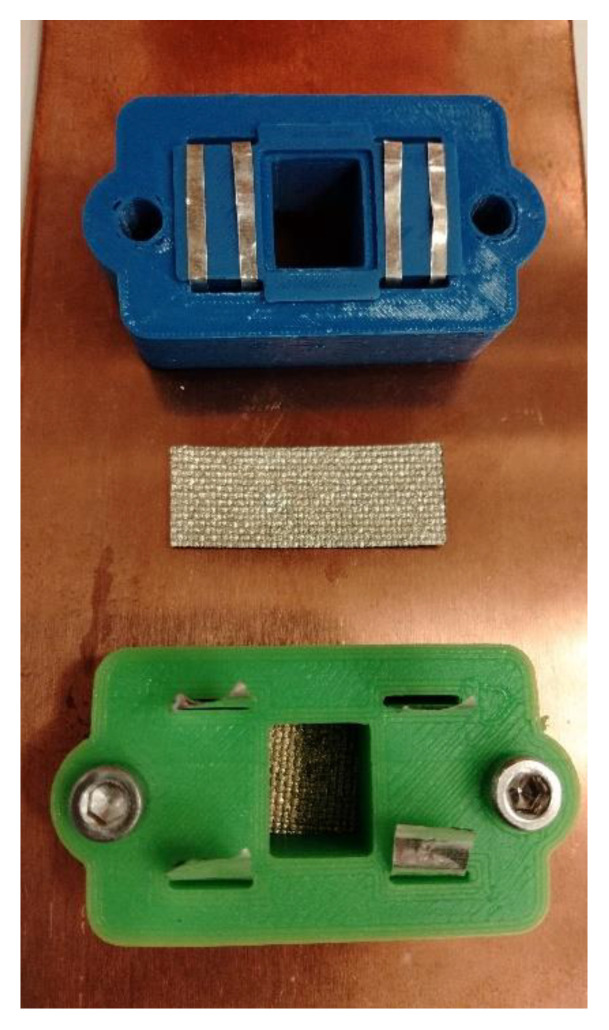
View of the sample in the holder mounted on the heat exchanger.

**Figure 3 materials-15-01814-f003:**
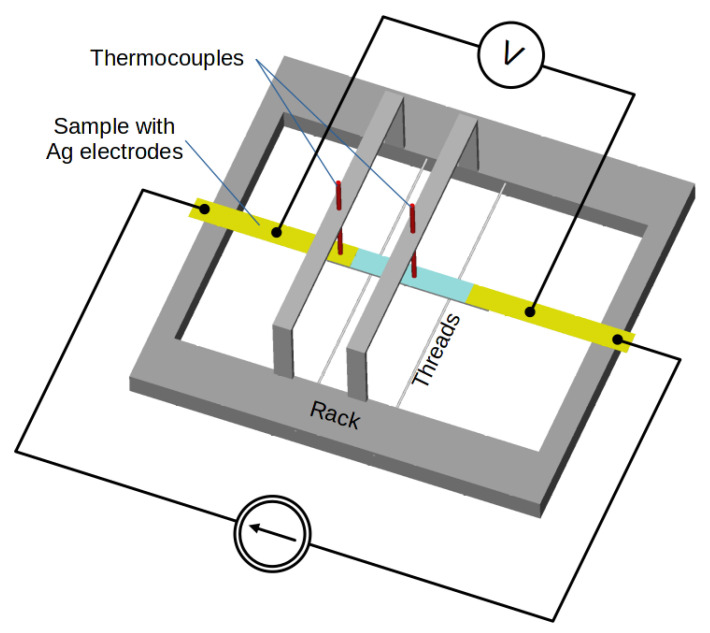
Schema of the rack with two thermocouples and thin threads supporting the sample.

**Figure 4 materials-15-01814-f004:**
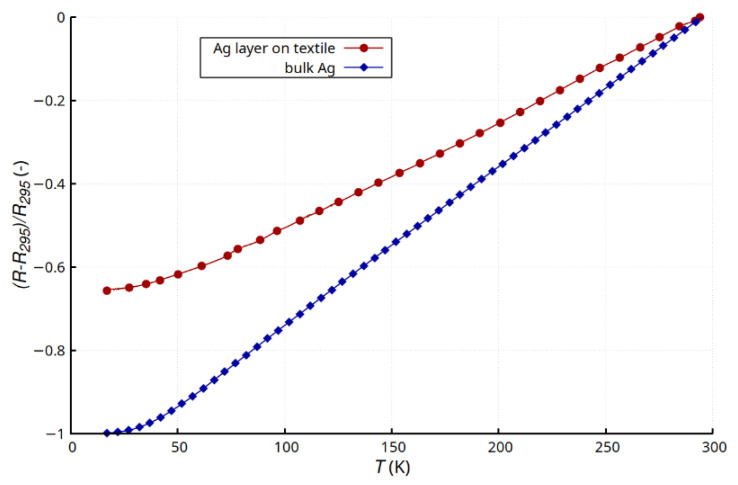
Comparison of relative changes in resistance and temperature of a thin Ag layer created on Cordura substrate and bulk Ag.

**Figure 5 materials-15-01814-f005:**
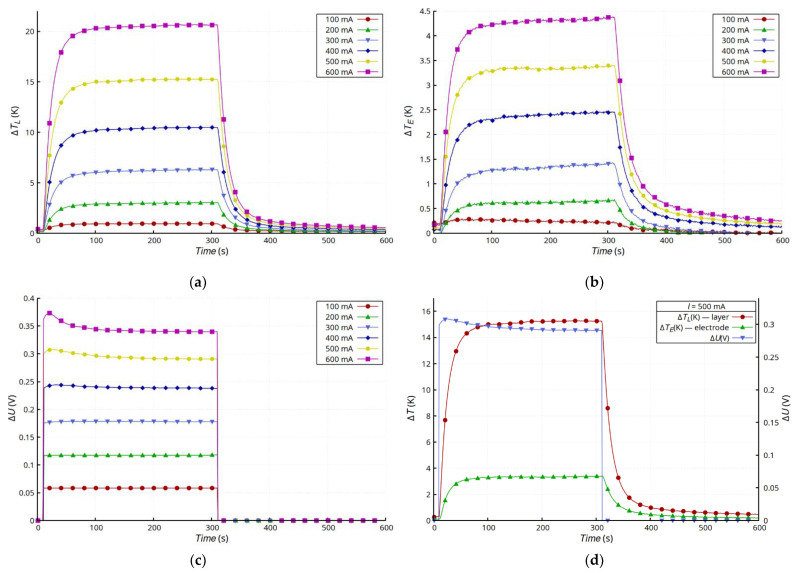
Results for G + T sample: time dependence of changes in temperature of Ag layer Δ*T_L_* (**a**), changes in temperature of electrode Δ*T_E_* (**b**), and drop voltage on sample Δ*U* (**c**), as well as comparison of results for *I* = 500 mA (**d**).

**Figure 6 materials-15-01814-f006:**
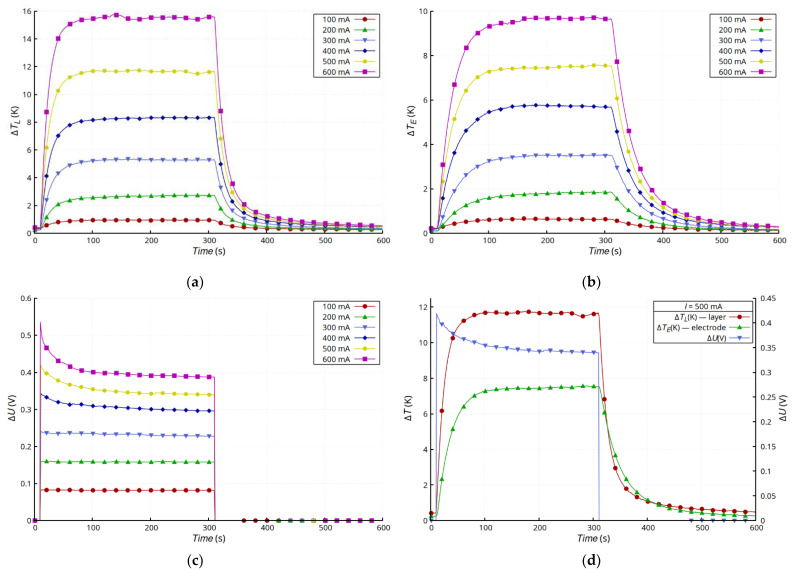
Results for C + T sample: time dependence of changes in temperature of Ag layer Δ*T_L_* (**a**), changes in temperature of electrode Δ*T_E_* (**b**), and drop voltage on sample Δ*U* (**c**), as well as comparison of results for *I* = 500 mA (**d**).

**Figure 7 materials-15-01814-f007:**
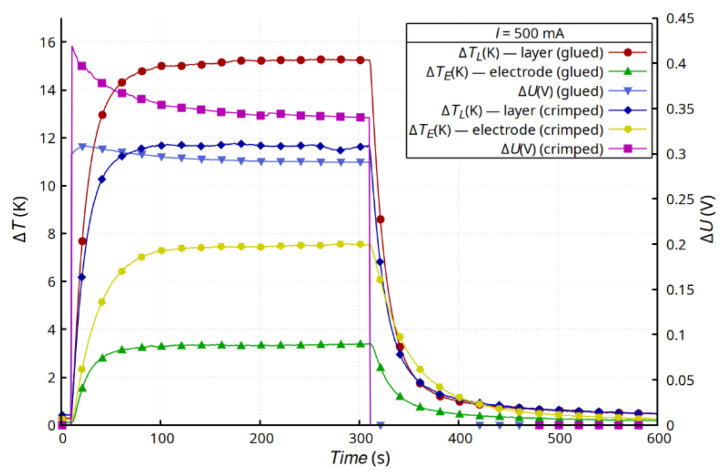
Comparison of results for the G + T sample and C + T sample for *I* = 500 mA.

**Figure 8 materials-15-01814-f008:**
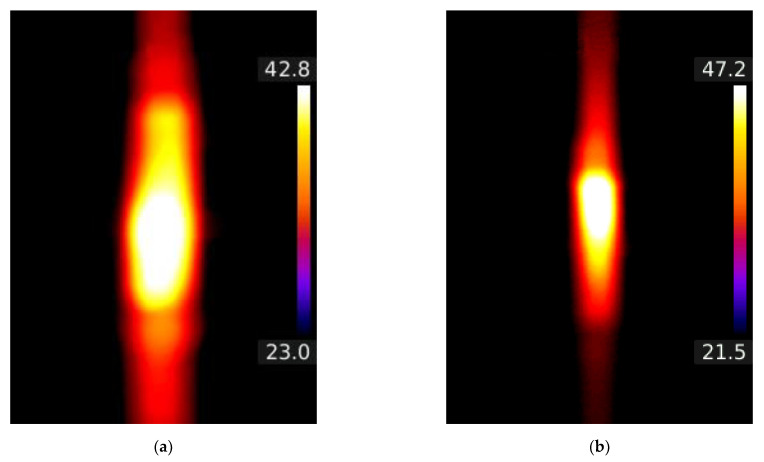
Thermographs for C + T sample (**a**) and G + T sample (**b**) for *I* = 600 mA.

**Figure 9 materials-15-01814-f009:**
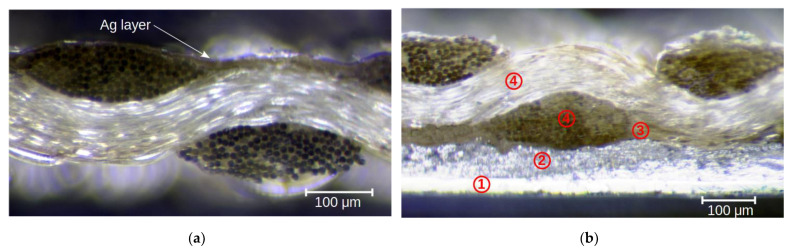
Microscope images of the cross-sections of a silver-coated Cordura textile (**a**) and a glued joint (**b**), where: 1—silver foil, 2—electroconductive paste, 3—polyurethane layer, 4—nylon threads.

**Figure 10 materials-15-01814-f010:**
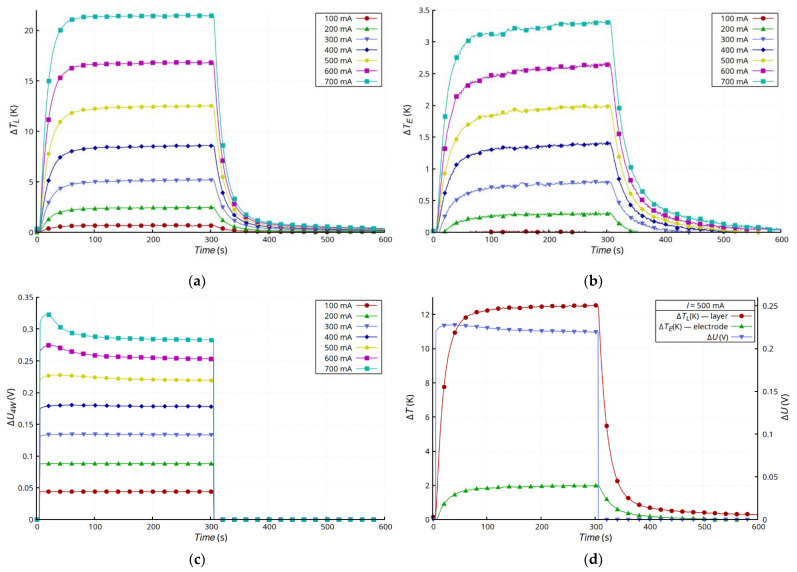
Results for sample M + T: time dependence of changes in temperature of Ag layer Δ*T_L_* (**a**); changes in temperature of electrode Δ*T_E_* (**b**); drop voltage on sample Δ*U_4W_* (**c**); comparison of results for *I* = 500 mA (**d**).

**Figure 11 materials-15-01814-f011:**
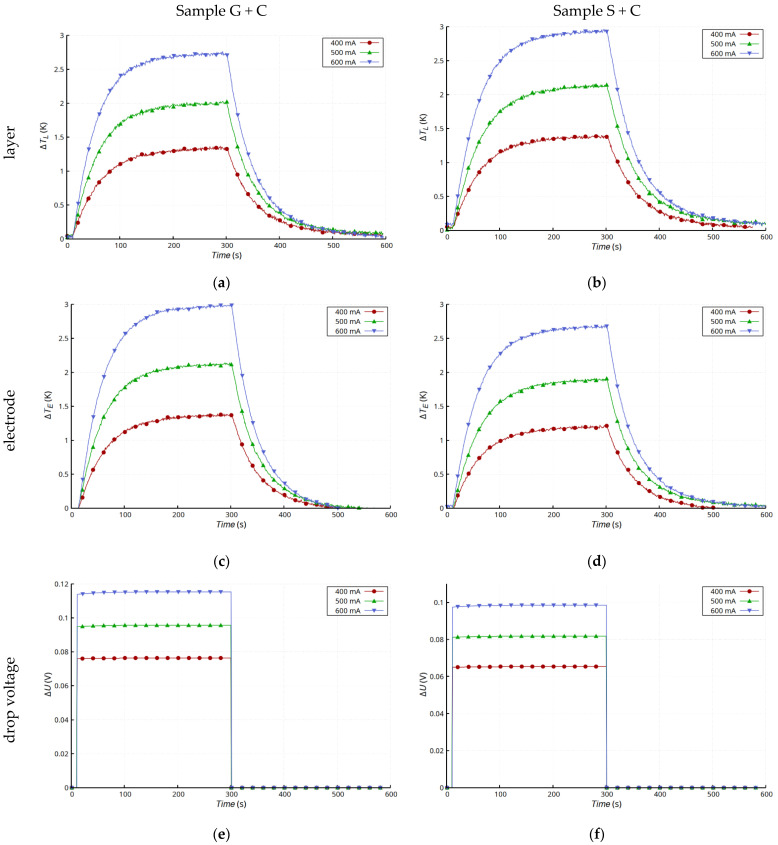
Comparison of results for glued electrodes (G + C) and soldered electrodes (S + C) on Al_2_O_3_ substrate: changes in temperature of Ag layer Δ*T_L_* (G + C) (**a**) and (S + C) (**b**); changes of temperature in electrode Δ*T_E_* (G + C) (**c**) and (S + C) (**d**); and drop voltage on samples *ΔU_4W_* (G + C) (**e**) and (S + C) (**f**).

**Table 1 materials-15-01814-t001:** Parameters of the samples.

Sample	S + C	G + C	G + T	C + T	M + T
Substrate	ceramic Al_2_O_3_	ceramic Al_2_O_3_	Cordura textile	Cordura textile	Cordura textile
Joining method	solder	glue	glue	rivet	glue
Leads	Ag strips 5 × 40 × 0.035 mm	Ag strips 5 × 40 × 0.035 mm	Ag strips 5 × 40 × 0.035 mm	Ag strips 5 × 40 × 0.035 mm	Ag strips 5 × 40 × 0.035 mm and Ag wires *φ* = 0.15 mm

## Data Availability

Not applicable.

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
