# Peer review of "Atypical Properties of a Thin Silver Layer Deposited on a Composite Textile Substrate"

_materials, 2022, doi:10.3390/ma15051814_

Round 1
Reviewer 1 Report
The manuscript presents interesting findings in the investigation of the electrical parameters of the thin layer formed on the textile composite surface. The investigations conducted by the authors are technically and scientifically interesting. Manuscript is written in clear style, but some points require additional information.
- The information about the composition of the electroconductive paste would be helpful.
Author Response
Dear Reviewer,
We would like to thank you very much for the time you spent correcting our paper and suggestions. They have helped us to improve the paper.
Below we have written answers to your comments. All of them are placed directly under the comments and they are highlighted in the text:
Reviewer #1:
The manuscript presents interesting findings in the investigation of the electrical parameters of the thin layer formed on the textile composite surface. The investigations conducted by the authors are technically and scientifically interesting. Manuscript is written in clear style, but some points require additional information.
The information about the composition of the electroconductive paste would be helpful.
We thank the reviewer for the valuable comment. According to the suggestion the following text has been placed in the manuscript: “The used glue consists of two components based on epoxy resin. After mixing the glue is the smooth paste of bright silver colour. More technical data is available at the producer website [https://amepox-mc.pl/elpox-kleje-epoksydowe/]”.
Attached please find the improved and upgraded text of our article.
Reviewer 2 Report
In the manuscript “Atypical properties of a thin silver layer deposited on a composite textile substrate” authors prepared 250-nm layer of Ag on textile and Al2O3, connected it with Ag foil strips by different methods. Electrical properties were tested as in a wide temperature range (15–295K), showed the R(T)”, however only “time dependence of temperature changes” were presented.
The manuscript is large, however not clear. The authors must give the correlation between mentioned 15–295K and presented Figures (connect it with current or ...).
Authors have shown 1) the results obtained for textile (untypical), and 2) the results obtained for Ag on Al2O3 (typical behavior) were represented. Probably opposite order (typical; untypical) will be more understandable for reader.
In addition:
- Please, rewrite Line 105 “However, it should be noted that when two conductive or semi-conductive materials are connected on one side, voltage appears on the unpaired terminals if the ends are kept at different temperatures.” Why Seebeck coefficient is important for thin layers deposited on textile?
- Details about CORDURA in Line 116.
- Line 118 “It is made of nylon threads covered with a polyurethane layer, thanks to which it is possible to create a thin electrically conductive layer on the surface.” Why polyurethane layer is very important? Why this textile was used? Are there types of textile without polyurethane layer?
- Table 1 is good but Where is description of G,T,C,M,S in (sample G+T); (sample C + T); (sample M + T); sample S + C?
- What electroconductive paste was used?
- Photo of “Ag layer to the ceramic substrate:...soldering (sample S+C)” (Line 153) is missing.
- Line 217 “The changes in resistance were typical for metals, where the phonon scattering mechanism predominates.” How it must look? Add Refs.
- Line 221 “temperature coefficient of resistance ..(αAg_layer = 2.6∙10-3 1/K)... coefficient for the bulk Ag (αAg_bulk = 3.8∙10-3 1/K)” How it was obtained? Or add Refs. How bulk Ag was obtained?
- Line 224 “structural defects”? Did the authors observed defects?
- 5: What happened after 300 sec for all samples?
- Line 246 “electrodes made of Ag foil”? there are only electrodes on textile and on Al2O3 in Table 1. Rewrite.
- Line 249 “The electrodes have much lower thermal conductivity” What electrodes and in comparison to what?
- Line 266 “Lower temperatures” or lower changing of temperature?
- Line 288 “The unusual changes in the resistance of the samples”. All samples? Please, describe in details What is “usual changes”, add Refs.
- Did the author studied pure textile substrate and Al2O3 in the same way?
Author Response
Dear Reviewer,
We would like to thank you very much for the time you spent correcting our paper and suggestions. They have helped us to improve the paper.
Below we have written answers to your comments. All of them are placed directly under the comments and they are highlighted in the text:
In the manuscript “Atypical properties of a thin silver layer deposited on a composite textile substrate” authors prepared 250-nm layer of Ag on textile and Al2O3, connected it with Ag foil strips by different methods. Electrical properties were tested as in a wide temperature range (15–295K), showed the R(T)”, however only “time dependence of temperature changes” were presented.
The manuscript is large, however not clear. The authors must give the correlation between mentioned 15–295K and presented Figures (connect it with current or ...).
We performed two types of thin Ag studies on textile: a wide temperature quasi-static study without Joule heating, and a time-dependent study with Joule heat generated in the thin layer by an electric current. In the first case, the sample was cooled with low dynamics of temperature changes, and the temperature gradients in the sample has been eliminated by using the high heat capacity of the copper exchanger. Figure 4 shows the relative changes of the resistance of the layer compared with a pure bulk silver sample for a temperature range of 15–295K. The aim of the study at cryogenic temperatures was to determine the nature of the dependence of the layer resistance on the temperature in comparison with the reference pure silver. In the second case, the sample was tested under natural convection and powered from a controlled current source. The aim of the research was to determine the dependence of the layer resistance on temperature when the layer is the Joule heat source. The results of the time dependence of temperature, current and voltage are shown in Figs. 5 and 6.
According to the suggestion, we have improved the Results and Discussion Section in the manuscript.
Authors have shown 1) the results obtained for textile (untypical), and 2) the results obtained for Ag on Al2O3 (typical behavior) were represented. Probably opposite order (typical; untypical) will be more understandable for reader.
Our aim is to investigate the behaviour of metallic layers made of silver on a composite textile substrate. The tests on Al2O3 are additional tests and constitute only a supplement to the research carried out in order to compare the behaviour of thin metallic structures and glued joints on substrates used, among others, in electronics. According to the authors the proposed order of presenting the results is more reasonable.
In addition:
-
- Please, rewrite Line 105 “However, it should be noted that when two conductive or semi-conductive materials are connected on one side, voltage appears on the unpaired terminals if the ends are kept at different temperatures.” Why Seebeck coefficient is important for thin layers deposited on textile?
This sentence is the supplementary one to the previous part of the Introduction where we describe the connections between the metallic wires and the aluminium substrate. The Seebeck effect is related to the difference in the work of electron output from materials, but it is a parasitic effect at crio temperatures, probably it occurs in our research but the applied measurement method (4-wire) and changing the direction of the measurement current eliminate the thermoelectric force from the results. We do not observe the Seebeck effect in our case.
Additional reference which treats about Seebeck in the case of silver nanostructures has been also added to the paper:
- Kockert M, Kojda D, Mitdank R, et al. Nanometrology: Absolute Seebeck coefficient of individual silver nanowires. Sci Rep. 2019;9(1):20265. Published 2019 Dec 30. doi:10.1038/s41598-019-56602-9
-
- Details about CORDURA in Line 116.
The information about the features of Cordura is placed in the text: “Its high mechanical strength justifies the choice Cordura as the base material. The area density of the material is 460 g/cm2. It is made of nylon threads covered with a polyurethane layer, thanks to which it is possible to create a thin electrically conductive layer on the surface”
-
- Line 118 “It is made of nylon threads covered with a polyurethane layer, thanks to which it is possible to create a thin electrically conductive layer on the surface.” Why polyurethane layer is very important? Why this textile was used? Are there types of textile without polyurethane layer?
Cordura was used for the tests due to the mechanical strength of this textile base, which can be combined with various elements of clothing.
The following sentence has been added to text: “Due to the three-dimensional nature of the fabric, the polyurethane layer is necessary to create a continuous electricoconductive layer. Irregularities, which are a natural element as a consequence of weaving the thread, must be leveled to ensure that the textile substrate can be evened out. It is not possible to form a continuous layer with good electrical properties without an intermediate layer such as a polyurethane layer. Additionally, polyurethane fills the spaces between the nylon fibers and limits the possibility of their movement during the material deformation.”
- Table 1 is good but Where is description of G,T,C,M,S in (sample G+T); (sample C + T); (sample M + T); sample S + C?
The description of the samples and the entering of the abbreviations are placed in the text. Lines 188, 189, 195, 196 and 200
- What electroconductive paste was used?
The description of the electroconductive paste ELPOX AX 15S (Amepox Ltd, Lodz,Poland) has been placed in the text “The used glue consists of two components based on epoxy resin. After mixing the glue is smooth paste of bright silver colour. More technical data is available at the producer website [https://amepox-mc.pl/elpox-kleje-epoksydowe/].
- Photo of “Ag layer to the ceramic substrate:...soldering (sample S+C)” (Line 153) is missing.
We thank the reviewer for the valuable comment but the photo of the sample with glued and soldered electrodes is indistinguishable in the macro scale, so placing the second photo without any difference is not undesirable. Therefore there is only one presenting the sample with soldered electrodes in the article.
- Line 217 “The changes in resistance were typical for metals, where the phonon scattering mechanism predominates.” How it must look? Add Refs.
According to the reviewer suggestion the additional reference has been added to the manuscript:
- Guénault A. M., MacDonald D. K. C.: Electron and Phonon Scattering Thermoelectricity in Potassium and Alloys at Very Low Temperatures; Proceedings of the Royal Society of London. Series A, Mathematical and Physical Sciences, Vol. 264, No. 1316 (Oct. 24, 1961), pp. 41-59 https://www.jstor.org/stable/2414160
- Matula, R.A. Electrical Resistivity of Copper, Gold, Palladium, and Silver. Journal of Physical and Chemical Reference Data, 1979, 8, 1147-1298. doi:10.1063/1.555614
- Line 221 “temperature coefficient of resistance ..(αAg_layer = 2.6∙10-3 1/K)... coefficient for the bulk Ag (αAg_bulk = 3.8∙10-3 1/K)” How it was obtained? Or add Refs. How bulk Ag was obtained?
The data mentioned in the article are material data used by simulation packages, e.g. Comsol as a material library in models.
According to the reviewer's suggestion the following references has been added to the manuscript:
- Matula, R.A. Electrical Resistivity of Copper, Gold, Palladium, and Silver. Journal of Physical and Chemical Reference Data, 1979, 8, 1147-1298. doi:10.1063/1.555614
- Smith DR, Fickett FR. Low-Temperature Properties of Silver. J Res Natl Inst Stand Technol. 1995;100(2):119-171. doi:10.6028/jres.100.012
- Nagata, S., Ogino, M., and Taniguchi, S., Electrical Resistivity of Thin Metal Films Vapor-Quenched at 77 K, Phys. Sol. (a) 102, 711 (1987).
- Goree, W. S., and Scott, T. A., Pressure Dependence of Electrical Conductivity of Metals at Low Temperatures; J. Phys. Solids 27, 835–848 (1966).”
- Line 224 “structural defects”? Did the authors observed defects?
The mention about structural defects is complementary to highlight the differences between the temperature coefficients of resistance.
In our work we do not investigate the structural defect, which appear in the process of producing a thin layer. In some papers there are more information on the process of its producing in the Introduction section.
According to the editor suggestion it is also supplemented by the Thorton model: “The method of forming a thin layer in the sputtering process and its morphology is described by Thornton []. In creating layers, four zones are distinguished depending on the temperature of the substrate to the temperature of the source of metal evaporation ratio. The layer morphology is as follows: porous columnar structure, porous grain structure and recrystalized grain structure. In the research, the essence of the pressure of working gas is pointed out from the perspective of the phenomenon of energy loss by atoms of evaporated and deposited material as a result of collision with gas molecules in the vacuum chamber during the layer formation process.
In the case of higher pressure of the working gas, the atoms of the deposited material elastically collide with the sputtering gas atoms, and as the consequence, the vapor atom mean-free path is nearly the same as the source-to-substrate distance. For that reason, the deposition flux tilt component increases.
In the case when the working gas pressure is high, the mean kinetic energy of condensing particles is relatively small, and the mobility of the evaporated atoms is reduced. These factors influence the morphology of the structure, which is described as Zone I - porous columnar structure. In the case of low pressure of the working gas, the mobility of the atom increases, and it leads to the formation of a dense structure called the Zone T. It is characterized by dense grain boundary arrays, high dislocation and less roughness.
The working gas pressure does not affect the structure of the thin film which can be characterized as porous grain structure - Zone II or recrystalized grain structure - Zone III.”
Additionally, the text has been placed in the paper: “Fig. 4 presents the nature of changes in R(T) for the layer created on textile substrate and the pure bulk Ag. They are very similar and a flattening of the characteristics below 40K is observed, it is equated with deactivation of the phonon mechanism. For a pure silver, the dynamics of changes is greater, which should be equated with a lower concentration of defects.”
The following paper has been added to the manuscript:
- Kluth O., Schöpe G., Hüpkes J., Agashe C., Müller J., Rech B., Modified Thornton model for magnetron sputtered zinc oxide: film structure and etching behavior, Thin Solid Films, 442, (1–2), 2003, 80-85, doi.org/10.1016/S0040-6090(03)00949-0.
- Thornton, J. A. Influence of apparatus geometry and deposition conditions on the structure and topography of thick sputtered coatings. Vac. Sci. Technol. 11, 666–670 (1974)
- Martin, P. M. Handbook of deposition technologies for films and coatings : science, applications and technology. (Elsevier, 2010)
- 5: What happened after 300 sec for all samples?
After 300 s the temperature of the samples decreased due to switching off the power source. It can be seen in all Figures presenting the obtained results (e.g. Fig 5).
- Line 246 “electrodes made of Ag foil”? there are only electrodes on textile and on Al2O3 in Table 1. Rewrite.
In our samples the connecting to the external system is made of silver strips, what is mentioned in the Table 1. So in this case we have changed the text “Ag foil” to “Ag strips”
- Line 249 “The electrodes have much lower thermal conductivity” What electrodes and in comparison to what?
We thank the reviewer for his thorough analysis of the text. There is our mistake. The correct sentence is as follows : ”The substrate has much lower thermal conductivity in comparison to electrodes”
- Line 266 “Lower temperatures” or lower changing of temperature?
Thank the reviewer for the suggestion. We have changed the information to "lower changing of temperature.
- Line 288 “The unusual changes in the resistance of the samples”. All samples? Please, describe in details What is “usual changes”, add Refs.
We observed the unusual changes in the resistance of thin layer created on textile substrate.
The sentence was expanded. “The unusual changes in the resistance of the layer created on the textile substrate were directly related to the structure of the composite substrate (compare the plots in Fig. 9 and 10). “
The information about the difference in behaviour of layers created on textile and ceramic substrate is placed in the text when the plots presented in Fig 9 and 10 are described.
- Matula, R.A. Electrical Resistivity of Copper, Gold, Palladium, and Silver. Journal of Physical and Chemical Reference Data, 1979, 8, 1147-1298. doi:10.1063/1.555614
- Did the author studied pure textile substrate and Al2O3 in the same way?
All tests for textile samples and Al2O3 were conducted in the same conditions and in the same way. The information is placed in text: “The silver layer was deposited on the textile or ceramic in a single sputtering process, ensuring the same thickness of the Ag layer.” The conditions of the experiment have not been changed during the research so for that reason, we have not put the information twice in the paper.
All lines have updated numbers.
Attached please find the improved and upgraded text of our article.
Reviewer 3 Report
The paper of Lebioda and Korzeniewska presents the electrical analysis of Cordura (composite) textile coated with approx. 250 nm of silver thin film. The main objective was to investigate the resistance of the silver layer at different textile temperature values. In general the paper is well written and the subject is interesting for the readers of the Materials MDPI journal. Some improvements are requested before paper acceptance:
(1) Abstract: Concern the "...electrical parameters of the layer formed...", it is important to cite these parameter in the abstract. Also, "The parameters of the substrate can significantly affect the electrical parameters...", please inform what are the parameters of the substrate?;
(2) Introduction: What was the main novelty of the present work in relation to current literature? Please inform this at the end of the introduction topic;
(3) Materials and Methods, line 134-135: Please inform the mean value uncertainty of the silver film thickness;
(4) The authors concluded that the polyurethane layer was the source of the anomalous changes in the strength of the silver layer. If you remove the polyurethane layer, what result is expected?
Author Response
Dear Reviewer,
We would like to thank you very much for time you spent correcting our paper and suggestions. They have helped us to improve the paper.
Below we have written answers to your comments. All of them are placed directly under the comments and they are highlighted in the text:
The paper of Lebioda and Korzeniewska presents the electrical analysis of Cordura (composite) textile coated with approx. 250 nm of silver thin film. The main objective was to investigate the resistance of the silver layer at different textile temperature values. In general the paper is well written and the subject is interesting for the readers of the Materials MDPI journal. Some improvements are requested before paper acceptance:
(1) Abstract: Concern the "...electrical parameters of the layer formed...", it is important to cite these parameters in the abstract. Also, "The parameters of the substrate can significantly affect the electrical parameters...", please inform what are the parameters of the substrate?;
We understand the reviewer’s comment but we measure the resistance of the layer and we are interested in an atypical change in the resistance value resulting from deformation of the intermediate layer. Writing the value of resistance or resistivity does not matter, because in the article we describe the relative resistance changes corresponding to voltage drop changes. The sentence “The parameters of the substrate can significantly affect the electrical parameters... is changed into “The roughness and the ability to deform under the influence of the heat of the substrate can significantly affect the electrical parameters...”
(2) Introduction: What was the main novelty of the present work in relation to current literature? Please inform this at the end of the introduction topic;
The additional information has been placed at the end of the Introduction section: “One of the trends observed in developing miniature electronics is wearable electronics with particular emphasis on textronics. Connecting electrical circuits with textiles is a technological challenge. During the production of thin-film structures on textile substrates and in application research, the authors observed unusual changes in the resistance of the produced layers, not previously described in the literature. This article deals with the influence of glued, riveted and soldered joints between thin-layer structures produced on a textile substrate, which can be easily combined with other textile products, and other elements of electronic systems. This problem has not been described in the literature so far. The article also presents the results of research on atypical changes in the resistance of thin-film structures produced on a composite textile substrate in comparison with the changes in such layers produced on a ceramic substrate.”
(3) Materials and Methods, line 134-135: Please inform the mean value uncertainty of the silver film thickness;
We thank the reviewer for valuable comment. We have added the following text to the manuscript: “This value is the average value of several measurements on a glass substrate and the difference of the obtained results for the glass substrate does not exceed 20nm. Direct measurement on a textile base is not possible due to the elasticity of the substrate and the contact measurement method.”
(4) The authors concluded that the polyurethane layer was the source of the anomalous changes in the strength of the silver layer. If you remove the polyurethane layer, what result is expected?
Cordura was used for the tests due to the mechanical strength of this textile base, which can be combined with various elements of clothing. Due to the three-dimensional nature of the fabric, the polyurethane layer is necessary to create a continuous electroconductive layer. Irregularities, which are a natural element as a consequence of weaving the thread, must be levelled to ensure that the textile substrate can be evened out. It is not possible to form a continuous film with good electrical properties without an intermediate layer such as a polyurethane one. Additionally, polyurethane fills the spaces between the nylon fibers and limits the possibility of their movement during the material deformation. - Such information is placed in the text in the Materials and Methods section.
Attached please find the improved and upgraded text of our article.
Round 2
Reviewer 2 Report
The manuscript is already improved but there are few moments more:
1. Line 154 “Due to the three-dimensional nature of the fabric, the polyurethane layer is necessary to create a continuous electroconductive layer.”
Polyurethane is insulator (according to available information in internet), thus the polyurethane layer can be use only to get a flat surface before Ag deposition (not “to create a continuous electroconductive layer”).
2. The meaning of “TL – layer temperature” (Line 183) is presented only in figure capture of Fig. 1 (not in the text).
Later, was written in Line 297: “Figure 5 shows the results of measurements of layer temperature changes (Fig. 5a)”.
Thus, the meaning of ΔTL (formula for calculation) is missing.
3. Describe please what values of temperature T correspond to the applied current I (I = 100-700 mA)?
Author Response
Dear Reviewer,
We would like to thank you very much for the time you spent for correcting our paper and suggestions. They have helped us to improve the paper.
Below we have written answers to your comments. All of them are placed directly under the comments and they are highlighted in the text:
The manuscript is already improved but there are few moments more:
- Line 154 “Due to the three-dimensional nature of the fabric, the polyurethane layer is necessary to create a continuous electroconductive layer.”
Polyurethane is insulator (according to available information in internet), thus the polyurethane layer can be use only to get a flat surface before Ag deposition (not “to create a continuous electroconductive layer”).
After rereading the paper, due to duplication of information, we decided to remove this sentence from the text to make the manuscript clearer for reading.
- The meaning of “TL – layer temperature” (Line 183) is presented only in figure capture of Fig. 1 (not in the text).
The following text has been placed in the manuscript: “In Fig. 1, there is information about the location of the thermocouples to measure the temperature of the sample. Here STE means the position of the electrode temperature sensor (or connections) and STL means the position of the layer temperature sensor.”
- Later, was written in Line 297: “Figure 5 shows the results of measurements of layer temperature changes (Fig. 5a)”. Thus, the meaning of ΔTL (formula for calculation) is missing. Describe please what values of temperature T correspond to the applied current I (I = 100-700 mA)?
According to the reviewer’s suggestion, the manuscript has been supplemented with the following text: “The initial temperature of the samples was 21-23 C (as it can be seen in the thermographs), It means that the temperature after heating the sample with Joule's heat changed in the range of 23oC-40oC depending on the value of the flowing current (100-700mA). The selection of the temperature range is not accidental and correlates with the typical differences between the temperature of the human body and the ambient temperature.”
Attached please find the improved and upgraded text of our article.
Reviewer 3 Report
The authors performed the requested corrections very well, it is recommended the publication of the article in current form.
Author Response
Dear Reviewer,
thank you very much for your positive opinion about our work. We make every effort to ensure that our work is at a high level.
Ewa Korzeniewska,
Marcin Lebioda